# Effects of Post-Labeling Delay on Magnetic Resonance Evaluation of Brain Tumor Blood Flow Using Arterial Spin Labeling

**Ryutaro Ukisu [1],\*, Yusuke Inoue [1], Hirofumi Hata [2], Yoshihito Tanaka [2] and Rie Iwasaki [1]**

[1] Department of Diagnostic Radiology, Kitasato University School of Medicine, Sagamihara 252-0374, Kanagawa, Japan

[2] Department of Radiology, Kitasato University Hospital, Sagamihara 252-0375, Kanagawa, Japan

\* Correspondence: uki35ryu@gmail.com

**Abstract:** We investigated the effect of post-labeling delay (PLD) on the evaluation of brain tumor blood flow using arterial spin labeling (ASL) magnetic resonance (MR) imaging to assess the need for imaging with two PLDs. Retrospective analysis was conducted on 63 adult patients with brain tumors who underwent contrast-enhanced MR imaging including ASL imaging with PLDs of both 1525 and 2525 ms on a 1.5 T or 3 T MR unit. Blood flow was estimated in the tumors and normal-appearing brain parenchyma, and tumor blood flow was normalized by parenchymal flow. Estimates of tumor blood flow, parenchymal flow, and normalized tumor flow showed no statistically significant differences between PLDs of 1525 and 2525 ms. Close correlations between different PLDs were found, with the closest correlation for normalized tumor flow. These results were similarly observed for the 1.5 T and 3 T units. The blood flow estimates obtained using ASL MR imaging in patients with brain tumors were highly concordant between PLDs of 1525 and 2525 ms, irrespective of the magnetic field strength. It is indicated that imaging with a single, standard PLD is acceptable for ASL assessment of brain tumor perfusion and that additional imaging with a long PLD is not required.

**Keywords:** magnetic resonance imaging; arterial spin labeling; blood flow; brain tumor; post-labeling delay



## 1. Introduction

Magnetic resonance (MR) imaging methods for the evaluation of cerebral blood flow (CBF) include dynamic susceptibility contrast (DSC) imaging, dynamic contrast enhanced (DCE) imaging, and arterial spin labeling (ASL) imaging. The bases of the DSC and DCE imaging methods are the T2\* and T1 shortening effects of gadolinium-based contrast agents, respectively. In contrast, ASL imaging utilizes arterial blood water labeled magnetically as an endogenous contrast agent and does not require the administration of an exogenous contrast agent [1–4]. Although both pseudocontinuous arterial spin-labeling (pCASL) and pulsed arterial spin-labeling (PASL) sequences are widely used clinically for magnetic labeling, a pCASL sequence provides a better signal-to-noise ratio and is recommended. Using this labeling technique, arterial blood water is labeled with a radiofrequency pulse in the neck, and the brain is imaged after a waiting time, called post-labeling delay (PLD). The influx of magnetically labeled water into the brain tissues causes MR signal changes, and CBF is quantitatively estimated from the changes. ASL MR imaging is a safe, convenient technique to estimate CBF and is suitable for repeated assessments. It has been used to evaluate various disorders, including cerebrovascular diseases, brain tumors, epilepsy, neurodegenerative diseases, and psychiatric disorders [1,5,6].

Because the magnitude and distribution of ASL signals depend on the PLD, the selection of this imaging parameter is essential in ASL MR imaging [1–5,7]. The arterial transit time (ATT) is the time interval for blood labeled in the neck to reach the brain.

In patients with steno-occlusive diseases, arterial blood may be supplied via collateral circulation, possibly resulting in localized prolongation of the ATT. When the PLD is short relative to the ATT, labeled blood water may not reach the brain tissues sufficiently at the time of data acquisition, which causes an underestimation of CBF. In such a case, labeled blood water may remain in the arteries near the brain surface and appear as focal or curvilinear high-flow areas. Prolongation of the PLD suppresses the influence of the ATT and may allow labeled blood water to enter the brain tissues. However, the use of a long PLD reduces the signal change due to the T1 relaxation of labeled water, resulting in degradation of CBF images. Therefore, when selecting the PLD, sufficient arrival of labeled blood water into the brain tissues and preservation of magnetic labeling should be taken into consideration.

The optimal PLD may vary depending on patient conditions and is difficult to predict. Instead of selecting of a single, optimal PLD, ASL imaging may be performed with two different PLDs in occlusive cerebrovascular diseases [8–10]. Furthermore, ASL imaging with multiple PLDs permits the estimation of CBF corrected for ATT, and has been shown to be more accurate than single-PLD imaging [11–13]. However, imaging with two or more PLDs prolongs the examination time and increases burden on the patients and imaging facility. Therefore, the use of a single PLD is recommended for standard clinical practice [2].

ASL MR imaging is used clinically to evaluate brain tumors such as glioblastomas and meningiomas, and its utility has been reported for grading [14–16], assessment of vascularity [17], and diagnosis of recurrence [18,19]. A single PLD, ranging from 1.5 s to 2 s, is applied to the evaluation of brain tumors, and the significance of ASL imaging with two or more PLDs has not been established. In our clinical practice, we selected 1525 ms as a standard PLD value according to the vendor's recommendation. Considering the possible influence of the PLD under pathologic conditions, we acquired ASL images with a PLD of 2525 ms in addition to those with a PLD of 1525 ms in patients with brain tumors as well as in those with steno-occlusive diseases of the major cerebral artery and those who exhibited obvious abnormalities on ASL images at 1525 ms. In this study, we compared brain tumor blood flow estimates obtained by ASL imaging with the two PLDs. This study aimed to determine whether imaging with two different PLDs is required when evaluating brain tumor perfusion.

## 2. Materials and Methods

### 2.1. Subjects

A total of 63 adult patients (32 men, 31 women: age $58.7 \pm 14.7$ years, mean $\pm$ standard deviation) with brain tumors who underwent contrast-enhanced MR imaging including ASL imaging with PLDs of both 1525 and 2525 ms were retrospectively analyzed. Among them, 25 and 38 patients were examined on 1.5 T and 3 T MR units, respectively. For the 1.5 T unit, the final diagnosis was meningioma in 18 patients, glioblastoma in 6, and brain metastasis in 2. For the 3 T unit, the final diagnosis was meningioma in 18 patients, glioblastoma in 12, astrocytoma in 2, brain metastasis in 1, central neurocytoma in 1, brain stem glioma in 1, malignant lymphoma in 1, hemangioblastoma in 1, and ganglioglioma in 1. The diagnoses were made based on the imaging findings and clinical courses in 29 patients with meningiomas and one with brain metastasis; otherwise, the pathological diagnosis was established. Kitasato University Medical Ethics Organization (Sagamihara, Japan) approved this study (B19-265), and the need for informed consent was waived.

### 2.2. Imaging Procedures

MR imaging was conducted on four clinical scanners provided by GE Healthcare (Milwaukee, WI, USA): two Signa HDxt 1.5 T scanners of the same model with a 12-channel head component of the head-neck-spine coil, a Discovery MR750 3.0 T scanner with a 12-channel head component of the head-neck-spine coil, and a Discovery MR750w 3.0 T with a 12-channel GEM coil head-neck unit. No specific criteria were applied for selecting a scanner for each patient, other than safety problems regarding the high magnetic field.

Before contrast injection, ASL imaging was performed using a pCASL technique, consisting of a pCASL labeling sequence and a three-dimensional spiral fast spin echo acquisition sequence, with background suppression and no vascular suppression. Two image sets were acquired in each examination with PLDs of 1525 and 2525 ms, in this order. The following imaging parameters were applied to all image sets: bandwidth, $\pm$ 62.5 kHz; field of view, 240 mm; slice number, 36; and slice thickness, 4 mm. Other imaging parameters are described in Table 1. The slice number was increased to 38 if needed. Proton attenuation images were obtained together with ASL perfusion images, and quantitative CBF images were produced from these images using the software installed on the MR scanner.

**Table 1.** ASL imaging parameters.

| Parameter | MR Unit | | | | | |
| --- | --- | --- | --- | --- | --- | --- |
| | **1.5 T** | | **3 T (750)** | | **3 T (750w)** | |
| | 1525 | 2525 | 1525 | 2525 | 1525 | 2525 |
| PLD (ms) | 1525 | 2525 | 1525 | 2525 | 1525 | 2525 |
| Number of arms | 4 | 4 | 4 | 4 | 4 | 4 |
| Number of excitations | 2 | 3 | 2 | 2 | 2 | 2 |
| Repetition time (ms) | 4629 | 5324 | 4632 | 5327 | 4640 | 5335 |
| Echo time (ms) | 10.5 | 10.5 | 10.5 | 10.5 | 10.7 | 10.7 |
| Flip angle (°) | 155 | 155 | 111 | 111 | 111 | 111 |
| Spatial resolution (mm) | 5.07 | 5.07 | 5.52 | 5.52 | 5.77 | 5.77 |
| Scan time (min:s) | 1:42 | 2:40 | 1:42 | 1:57 | 1:42 | 1:57 |

The 1.5 T, 3 T (750), and 3 T (750w) denote Signa HDxt 1.5 T, Discovery MR750 3.0 T, and Discovery MR750w 3.0 T scanners, respectively. The parameters for acquiring 36 slices are presented, and the parameters for acquiring 38 slices are minimally different.

### 2.3. Data Analysis

Regions of interest (ROIs) were placed for the brain tumor and normal-appearing brain parenchyma on the CBF images using a workstation (Advantage Workstation, GE Healthcare), and mean blood flow values in the ROIs were determined in the unit of mL/min/100 g. One lesion was analyzed for each patient. Postcontrast T1-weighted images, fluid attenuation inversion recovery (FLAIR) images, and CBF images at 1525 and 2525 ms were displayed on the workstation. Elliptical ROIs were set in the solid, presumably viable part of the tumor. The area of the tumor ROI was at least 30 mm$^2$, and two or more ROIs were set for one lesion when central necrosis prevented setting an ROI of adequate size. The ROI for the normal-appearing parenchyma, of at least 60 mm$^2$ in area, was placed in the corresponding contralateral region on the same image slice. The parenchymal ROI was placed in the adjacent intact region when the lesion was located centrally. Identical ROIs were applied to the images at 1525 and 2525 ms. Normalized tumor blood flow was calculated as the ratio of tumor blood flow to parenchymal flow. Tumor flow, parenchymal flow, and normalized tumor flow were compared between different field strengths and different PLDs. Comparisons between different PLDs were also made in subgroups created according to the pathology (meningiomas and glioblastomas), gender (men and women), or age (younger than 70 years and equal to or older than 70 years).

### 2.4. Statistical Analysis

Data are presented as means $\pm$ standard deviations. Comparisons between different PLDs were made using a paired *t* test, and those between different field strengths were made using Welch's *t* test. Linear regression analysis was performed by the least squares method. A *p* value less than 0.05 was deemed statistically significant.

### 3. Results

The CBF images obtained at PLDs of 1525 and 2525 ms are exemplified in Figure 1. When mean values of blood flow estimates were compared, neither tumor blood flow, parenchymal blood flow, nor normalized tumor blood flow differed significantly between

1.5 T and 3 T (Figure 2). Moreover, no significant differences were found between PLDs of 1525 and 2525 ms in these blood flow estimates, irrespective of the data analyzed, i.e., data of 1.5 T, those of 3 T, or all data.

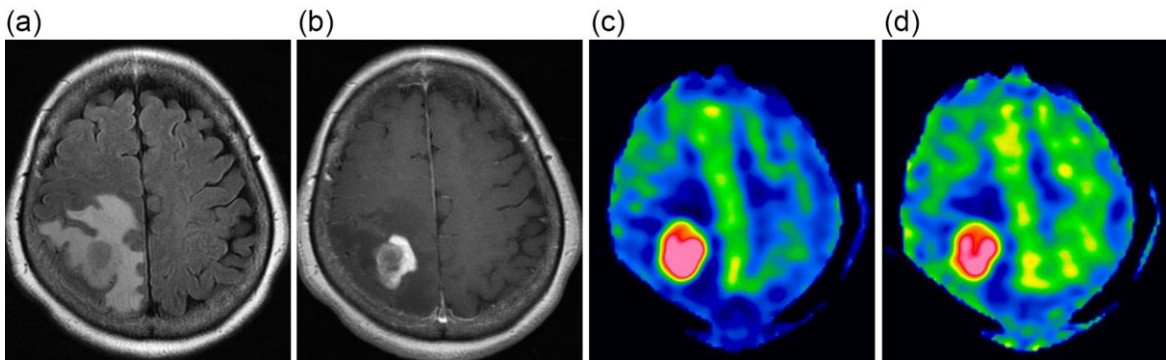

**Figure 1.** A 63-year-old female with glioblastoma. FLAIR (**a**) and contrast-enhanced T1-weighted (**b**) images show a heterogeneously enhanced mass lesion surrounded by extensive peritumoral edema. CBF images at 1525 ms (**c**) and 2525 ms (**d**), presented using the same display window, demonstrate high blood flow in the lesion.

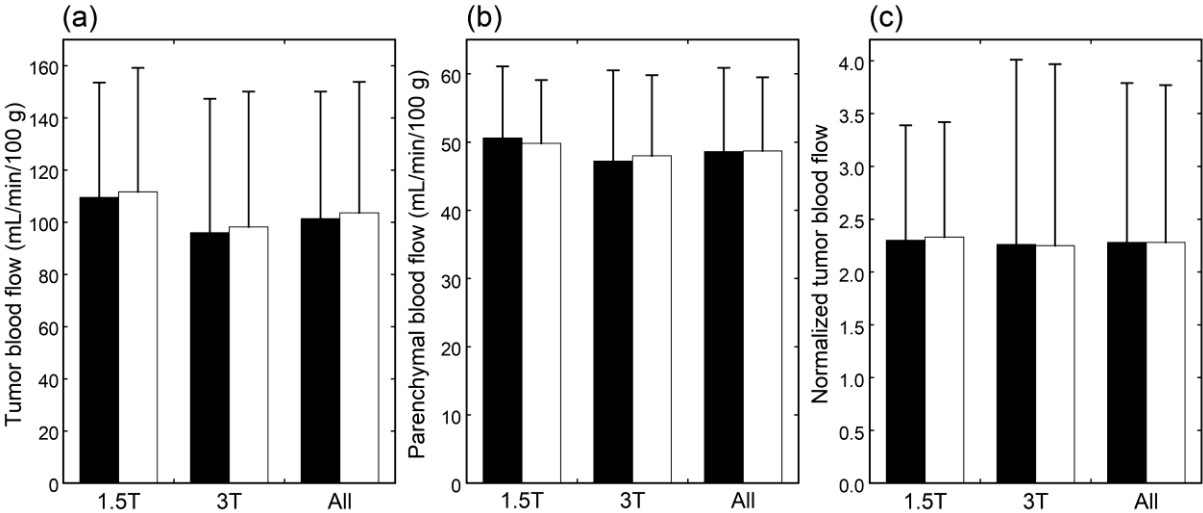

**Figure 2.** Tumor blood flow (**a**), parenchymal blood flow (**b**), and normalized tumor blood flow (**c**) in patients imaged at 1.5 T (*n* = 25), those imaged at 3 T (*n* = 38), and all patients (*n* = 63). The black and white bars indicate mean values estimated with PLDs of 1525 and 2525 ms, respectively. Error bars represent standard deviations.

Irrespective of the data analyzed, tumor blood flow (Figure 3), parenchymal blood flow (Figure 4), and normalized tumor blood flow (Figure 5) showed close correlations between different PLDs, with the closest correlation for normalized tumor flow. Although one outlier was found for normalized tumor flow, its exclusion did not affect the correlation substantially. The correlation coefficients between different PLDs were similar between 1.5 T and 3 T.

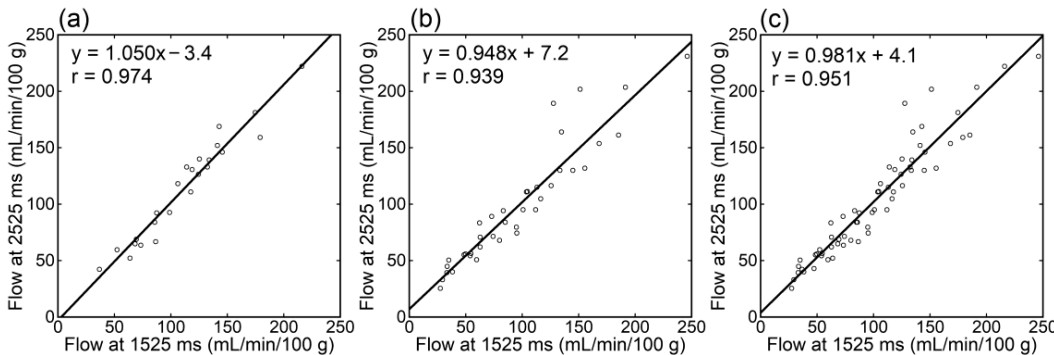

**Figure 3.** Relationships of tumor blood flow estimates between different PLDs in patients imaged at 1.5 T (**a**), those imaged at 3 T (**b**), and all patients (**c**).

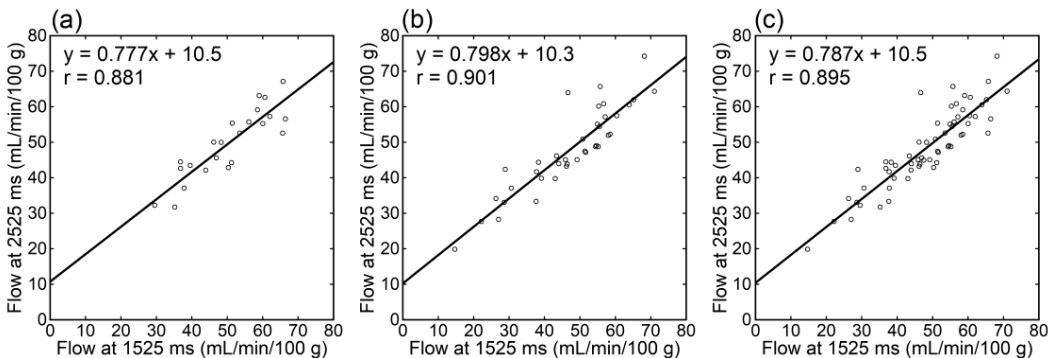

**Figure 4.** Relationships of parenchymal blood flow estimates between different PLDs in patients imaged at 1.5 T (**a**), those imaged at 3 T (**b**), and all patients (**c**).

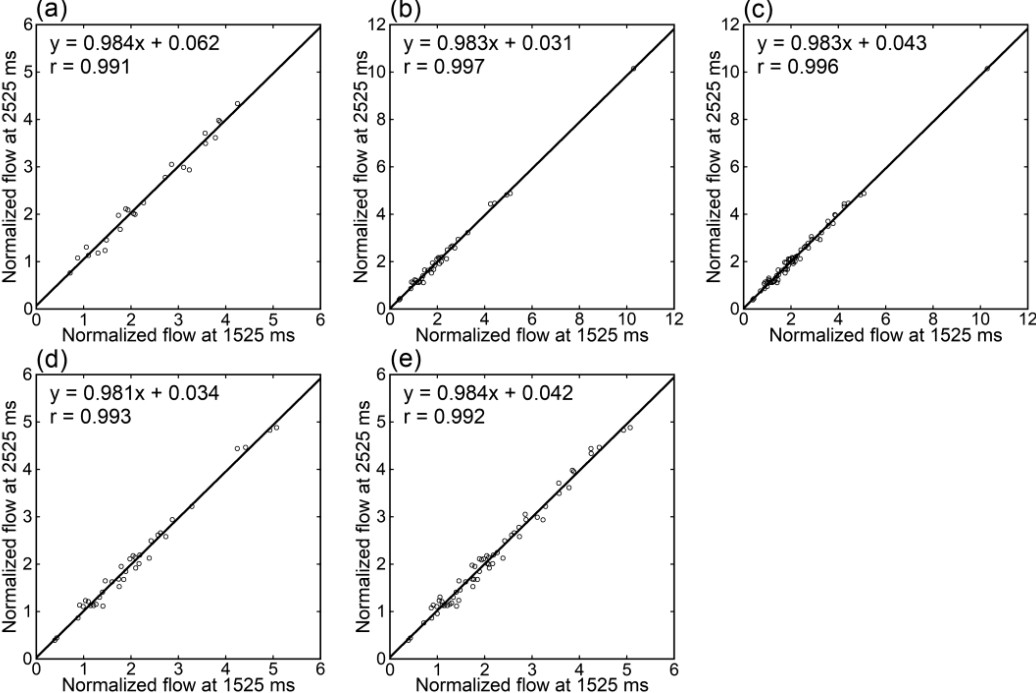

**Figure 5.** Relationships of normalized tumor blood flow estimates between different PLDs in patients imaged at 1.5 T (**a**), those imaged at 3 T (**b**,**d**), and those imaged at either 1.5 T or 3 T (**c**,**e**). One patient showing an exceptionally high value was excluded from panels (**d**,**e**).

In the meningioma subgroup, the comparison between tumor blood flow estimates obtained at 1525 and 2525 ms showed no significant difference and a close correlation (Table 2). Similar results were demonstrated in the glioblastoma subgroup, gender-based subgroups, and age-based subgroups.

**Table 2.** Tumor blood flow in the subgroups.

| Group | Tumor Blood Flow(mL/min/100 g) | | *r* | *n* |
|---|---|---|---|---|
| | **1525 ms** | **2525 ms** | | |
| Meningioma | 109.6 ± 50.5 | 112.4 ± 50.1 | 0.945 | 36 |
| Glioblastoma | 100.3 ± 44.6 | 99.3 ± 44.8 | 0.970 | 18 |
| Men | 110.8 ±49.4 | 116.4 ±54.4 | 0.948 | 32 |
| Women | 91.7 ± 46.8 | 90.4 ± 42.4 | 0.966 | 31 |
| <70 years | 99.7 ± 51.0 | 98.8 ± 44.7 | 0.976 | 46 |
| ≥70 years | 106.0 ± 42.8 | 108.2 ± 55.3 | 0.918 | 17 |

Mean ± SD is presented.

## 4. Discussion

In this study, we retrospectively analyzed ASL MR images in patients with brain tumors to evaluate the effect of the PLD. The estimates of tumor blood flow, normal-appearing parenchymal blood flow, and normalized tumor blood flow showed no statistically significant differences and exhibited high correlations between PLDs of 1525 and 2525 ms. Such high concordances were demonstrated irrespective of the magnetic field strength, tumor pathology, gender, or age. The results of this study indicate that the influence of PLD selection can be neglected in the evaluation of brain tumors and that imaging with two different PLDs is not required.

The evaluation of blood flow in brain tumors is one of the major roles of ASL MR imaging. Meta-analysis studies indicated the usefulness of ASL imaging for grading gliomas [14,15]. Batalov et al. [20] analyzed 253 patients with gliomas who underwent ASL imaging with a PLD of 1525 ms. They demonstrated higher tumor blood flow in high-grade gliomas than in low-grade gliomas. The sensitivity and specificity were high in differentiating between high-grade and low-grade gliomas, although they were low in differentiating between grade III and grade IV gliomas. Detection of recurrent or residual lesions is another important role of ASL imaging in brain tumors [18,19]. Cohen et al. [21] investigated the diagnostic performance of ASL imaging for the detection of postoperative residual lesions in 75 patients with glial or metastatic tumors. They assessed tumor perfusion preoperatively using DSC imaging and residual lesions within 72 h after surgery using ASL imaging with a PLD of 2025 ms. The diagnostic performance for the detection of residual lesions was high for lesions where hyperperfusion was shown preoperatively. Meningiomas are often highly vascularized, and arterial embolization may be performed preoperatively. Koizumi et al. [17] performed ASL imaging with a PLD of 1525 ms in 25 patients with meningiomas before surgery, and demonstrated higher blood flow in angiomatous meningiomas determined histopathologically than in non-angiomatous meningiomas. They also suggested the usefulness of ASL evaluation after preoperative embolization of meningiomas [22].

ASL imaging is commonly performed with a single PLD, and the selection of the PLD value is essential for reliable measurement. Prolongation of the PLD is useful to avoid underestimation of blood flow due to delayed arrival into the target tissues; however, the signal-to-noise ratio decreases due to the loss of magnetic labeling. The PLD should be selected based on the balance between arrival into the tissues and preservation of magnetic labeling. The PLD values described in the previous studies, including the brain tumor studies mentioned above [17,20,21], were in the range of 1.5 s to 2 s. A consensus statement recommended 2000 ms for adult clinical patients, considering possible prolongation of the ATT under pathological conditions [2].

Previous studies in healthy volunteers demonstrated no significant systematic differences in CBF estimates between different PLDs [23,24]. However, the PLD selection may influence blood flow estimates obtained by ASL MR imaging under unusual conditions. Acetazolamide dilates cerebral vessels, and consequently, increases CBF. In healthy subjects injected with acetazolamide, CBF estimates were shown to be higher with a short PLD of 1525 ms than with a PLD of 2525 ms, presumably due to the shortening of the ATT [23].

Localized prolongation of the ATT may occur in occlusive cerebrovascular diseases, leading to underestimation of CBF in the affected regions using ASL imaging with a standard PLD. Apparent hypoperfusion on ASL images obtained with a standard PLD may represent prolonged ATT but not decreased CBF. To discriminate between prolonged ATT and decreased CBF, ASL imaging with a long PLD may be added to standard ASL imaging. Haga et al. [8] studied ten patients with unilateral stenosis or occlusion of the internal carotid artery or middle cerebral artery using a dual PLD method of ASL imaging with two PLDs (1525 and 2525 ms). In five patients, CBF in the territory of the middle cerebral artery on the affected side was low in the images at 1525 ms and improved in those at 2525 ms. Acetazolamide loading single-photon emission tomography demonstrated a corresponding decrease in cerebrovascular reserve. Lyu et al. [9] performed ASL imaging with two PLDs of 1.5 and 2.5 s in 41 patients with middle cerebral artery stenosis. Mean CBF was significantly lower on the lesion side than on the normal side at 1.5 s; however, such a difference was not observed at 2.5 s, presumably due to collateral perfusion to the affected region. Collateral flow assessed based on the comparison between CBF estimates at 1.5 and 2.5 s correlated with the grade of collateral flow assessed on conventional angiography.

In our clinical protocol for ASL imaging, the standard PLD value was set at 1525 ms, and imaging with a PLD of 2525 ms was added under some pathologic conditions including brain tumors. In this study, we compared blood flow estimates obtained with two PLDs of 1525 and 2525 ms in brain tumor patients and demonstrated a high concordance between the PLDs. The results of this study indicate that ASL MR imaging with a single, standard PLD is acceptable for the evaluation of brain tumor perfusion, and the addition of imaging with a longer PLD is not required. The concordance between estimates at 1525 and 2525 ms indicates that prolonged ATT does not cause problems in ASL evaluation of brain tumor blood flow. Therefore, the shorter PLD is preferable to the longer PLD because of a higher signal-to-noise ratio. The PLD value of 2000 ms recommended in the consensus statement [2] should also be acceptable for brain tumor evaluation.

The magnetic field strength is an essential factor in determining the quality of MR imaging. The use of a 3 T MR scanner is recommended for ASL imaging [2]. The signal-to-noise ratio is lower at 1.5 T than at 3 T in general, and the difference is enhanced for ASL imaging because magnetic labeling is lost more rapidly due to a shorter T1 relaxation time at 1.5 T. In this study, the number of excitations was increased in imaging with a PLD of 2525 ms at 1.5 T to compensate for decreased signals. The concordance of blood flow estimates between PLDs of 1525 and 2525 ms was shown not only at 3 T but also at 1.5 T. Imaging with a single PLD appears to be acceptable for the evaluation of brain tumor perfusion irrespective of the field strength. Blood flow estimates were similar between patients examined on 1.5 T and 3 T scanners. This comparison gives insight into the characteristics of the study subjects; however, the similarity does not mean a concordance in blood flow measurement between 1.5 T and 3 T. A volunteer study indicated a weaker correlation between different scanner models than between different scanners of the same model [24]. The use of the same scanner or the same scanner model is desired for the longitudinal assessment of a given patient.

The study subjects included brain tumor patients with various pathologies: many patients with meningiomas and glioblastomas and a small number of patients with other pathologies. The blood supply pathway differs between meningiomas and glioblastomas; meningiomas are supplied by the meningeal and cerebral arteries. We analyzed the patients with meningiomas and glioblastomas separately and found a high concordance between PLDs of 1525 and 2525 ms for each pathology. The results support the appropriateness of a

single PLD for imaging meningiomas and glioblastomas. Future studies regarding other pathologies are desired.

Because prolongation of the ATT is expected in older patients, application of a longer PLD is recommended for healthy subjects older than 70 years (2000 ms) than for those younger than 70 years (1800 ms) [2]. Because of the difficulty of expectation of the ATT in patients with various clinical conditions, the longer PLD (2000 ms) is recommended for clinical patients. In this study, a high concordance in tumor blood flow estimates between PLDs of 1525 and 2525 ms was demonstrated in patients equal to or older than 70 years, and the need for imaging with a long PLD was not indicated. Additionally, a good concordance between the two PLDs was also found for both men and women, indicating that considering gender is not required in selecting the PLD for the evaluation of brain tumor perfusion.

The number of arms is an imaging parameter that adjusts the spatial resolution in ASL imaging: an increased number of arms leads to higher spatial resolution at the expense of a prolonged scan time. Although the typical arm number is eight, four arms were used in this study to avoid prolongation of the examination time due to imaging with two different PLDs, which lowered spatial resolution. Because of the observed absence of significant effects of the PLD, we now acquire ASL images in brain tumor patients with eight arms and a PLD of 1525 ms.

Although mean tumor blood flow was almost identical between the two PLDs, some random variations in tumor flow estimates were observed and reduced by normalization by parenchymal flow estimates, as indicated by the improved correlation. In MR imaging, scanner tuning, consisting of shimming of the magnetic field, adjustment of the transmit gain, adjustment of the receive gain, and determination of the resonant frequency, is performed prior to each image acquisition. Scanner tuning was indicated to be a major source of random variations in CBF measurements performed serially in a single imaging session using the same imaging parameters in healthy subjects [23]. Serial imaging of healthy subjects with different parameters demonstrated larger random variations between different PLDs than between different numbers of arms, possibly due to individual differences in the ATT [24]. The results of this study suggest that such random variations occur similarly in blood flow estimates for the tumor and normal-appearing parenchyma and thus are reduced by the normalization of tumor flow by parenchymal flow. Although the setting of the ROI for the control region may introduce additional variations, normalized tumor blood flow may be a better marker than absolute tumor blood flow from the aspect of repeatability and may be helpful for longitudinal observation of a given patient.

There are limitations to this study and matters for future investigation. In this study, a pCASL sequence was used for the magnetic labeling of blood water. Although a pCASL sequence is preferable and commonly used, a PASL sequence may also be used in routine clinical practice [1,2]. Assessment of the effect of the PLD on imaging using a PASL sequence is beyond the scope of this study. We did not evaluate vascular diseases in this study. In patients with both brain tumors and vascular diseases, ASL imaging at short and long PLDs may be helpful. The study subjects included many patients with meningiomas and glioblastomas, and the number of patients with other pathologies was small. Additionally, most lesions showed high flow. Future studies enrolling patients with various pathologies and low-flow lesions are desired.

## 5. Conclusions

In this study, we evaluated the effect of PLD selection on estimating blood flow in brain tumors using ASL MR imaging. The blood flow estimates were highly concordant between PLDs of 1525 and 2525 ms irrespective of the magnetic field strength, which indicates that imaging with a single, standard PLD is acceptable for assessing brain tumor perfusion and that additional imaging with a long PLD is not required.

**Author Contributions:** Conceptualization, R.U. and Y.I.; methodology, R.U., Y.I., H.H. and Y.T.; formal analysis, R.U. and H.H.; investigation, R.U., Y.I., H.H. and R.I.; resources, Y.I.; data curation, R.U., Y.I. and H.H.; writing—original draft preparation, R.U. and Y.I.; writing—review and editing, R.U., Y.I., H.H., Y.T. and R.I.; visualization, R.U. and Y.I.; supervision, Y.I.; project administration, Y.I. All authors have read and agreed to the published version of the manuscript.

**Funding:** This research received no external funding.

**Institutional Review Board Statement:** This study was conducted in accordance with the Declaration of Helsinki, and approved by Kitasato University Medical Ethics Organization (B19-265).

**Informed Consent Statement:** The need for informed consent was waived based on the retrospective study design.

**Data Availability Statement:** The data are available upon reasonable request from the corresponding author.

**Conflicts of Interest:** The authors declare no conflict of interest.

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
