# Peer review of "Effects of Post-Labeling Delay on Magnetic Resonance Evaluation of Brain Tumor Blood Flow Using Arterial Spin Labeling"

_tomography, doi:10.3390/tomography9010036_

Round 1

Reviewer 1 Report

1.     The abstract lacks innovation and appears more like a synopsis. The writers must make it apparent how their work helps close the gap between published work and emerging trends. The research’s objective, key findings, and main conclusions should all be briefly stated in the abstract. It must be able to stand alone since an abstract is frequently offered apart from the article.

2.     A little more detailed explanation of Magnetic resonance (MR) imaging is expected

3.     It is advised to outline the article’s structure towards the conclusion of the introduction. Contrasting the current study’s findings with some earlier, related research is advised.

4.     The authors are advised to check the grammatical errors throughout the manuscript.

5.     More recent studies need to be referred to improve the quality of the manuscript

6.     Please ensure your discussion section underscores the scientific value-added of your paper and the applicability of your findings/results. Please revise your discussion part in more detail. It would be best if you enhanced your contributions and limitations, underscored the scientific value-added of your paper, and/or the applicability of your findings/results and future study in this session.

7.     The conclusion section does not correctly address the critical points and future perspectives. The author should add a few essential points.

8.     The study should mention how this work can directly benefit society and what new measures need to be taken in order to address the challenges associated with the study

Author Response

Comment 1

The abstract lacks innovation and appears more like a synopsis. The writers must make it apparent how their work helps close the gap between published work and emerging trends. The research’s objective, key findings, and main conclusions should all be briefly stated in the abstract. It must be able to stand alone since an abstract is frequently offered apart from the article.

Reply 1

We revised the abstract to emphasize the objective and main conclusion, considering the limit of 200 words.

Comment 2

A little more detailed explanation of Magnetic resonance (MR) imaging is expected

Reply 2

We added some explanation about MR perfusion imaging and ASL techniques to the 1st paragraph of the Introduction section.

Comment 3

It is advised to outline the article’s structure towards the conclusion of the introduction. Contrasting the current study’s findings with some earlier, related research is advised.

Reply 3

According to the reviewer’s comment, we added descriptions related to comparisons with preceding studies in the 3rd and 4th paragraphs of the Introduction section and in the 2nd, 3rd, and 5th paragraphs of the Discussion section.

Comment 4

The authors are advised to check the grammatical errors throughout the manuscript.

Reply 4

We checked our revised manuscript for grammatical errors.

Comment 5

More recent studies need to be referred to improve the quality of the manuscript

Reply 5

We added eight references, including four papers published in 2020 or later.

Comment 6

Please ensure your discussion section underscores the scientific value-added of your paper and the applicability of your findings/results. Please revise your discussion part in more detail. It would be best if you enhanced your contributions and limitations, underscored the scientific value-added of your paper, and/or the applicability of your findings/results and future study in this session.

Reply 6

We revised the Discussion section, especially the 1st-6th paragraphs, to clarify the significance of our study.

We described the limitations and subjects for future investigation in the last paragraph of the original Discussion section. We explicitly stated the role of the paragraph in the revised manuscript.

Comment 7

The conclusion section does not correctly address the critical points and future perspectives. The author should add a few essential points.

Reply 7

We added a comment to emphasize the key message “additional imaging with a long PLD is not required.”

Comment 8

The study should mention how this work can directly benefit society and what new measures need to be taken in order to address the challenges associated with the study

Reply 8

We revised the manuscript to clarify the significance of our study.

Reviewer 2 Report

This retrospective study analyzed pCASL MRI in patients with brain tumors to evaluate the effect of two sets of the PLD, and the authors concluded that there's no difference between 1525ms and 2525ms and imaging with a single PLD is acceptable for brain tumor perfusion evaluation. 

The manuscript is well-written and presented clearly. But if there are more patients with different pathologies could be enrolled will give more convincing information.

If imaging with a single PLD appears to be acceptable for the evaluation of brain tumor perfusion irrespective of the field strength. Could it be concluded that which one (1525ms vs. 2525ms) is the better one for single scan? Are there any other shorter or longer better PLD lengthes for ASL for certain pathology?

Figure 2.  Please put the sample number in the legends to show how many patients here.

Author Response

Comment  1

The manuscript is well-written and presented clearly. But if there are more patients with different pathologies could be enrolled will give more convincing information.

Reply 1

Considering the possibility of pathology-type dependence, we added separate analysis for meningiomas and glioblastomas and related discussion (the 8th paragraph of the Discussion section). Need for further investigation regarding various tumor pathologies was commented in the last paragraph of the original Discussion section, and was also commented in the 8th paragraph of the revised Discussion section.

Comment 2

If imaging with a single PLD appears to be acceptable for the evaluation of brain tumor perfusion irrespective of the field strength. Could it be concluded that which one (1525ms vs. 2525ms) is the better one for single scan? Are there any other shorter or longer better PLD lengthes for ASL for certain pathology?

Reply 2

The shorter PLD, 1525 ms, should be better because of a higher signal-to-noise ratio. We stated this recommendation in the 6th paragraph of the revised Discussion section.

The PLD used commonly is in the range of 1.5 s and 2 s, and a PLD of 2.5 s is applied additionally when appropriate. We have no idea regarding a PLD shorter than 1.5 s or longer than 2.5 s.

Comment 3

Figure 2.  Please put the sample number in the legends to show how many patients here.

Reply 3

We revised the legend according to the reviewer’s comment.

Reviewer 3 Report

Major Comments:

1.   Please indicate the importance of the study in the Introduction?

2.   Why did the authors select specific PLDs of 1525 and 2525 ms in their study?

3.   Why didn’t the authors provide ASL imaging parameters gender wise? Was there any sex-dependent variation in these parameters? The authors should provide gender-wise data.

4.   The authors consider the data from two different scanner in their study. Can the authors eliminate potential differences due to different scanning machine?

5.   Did the authors find any differences in the ASL assessment depending upon advancement of the tumor?

 Minor Comments:

1.     In the figure 2, error bars are too high!

2.     The manuscript should be checked for the grammatical errors and typos.

Author Response

Comment 1

Please indicate the importance of the study in the Introduction?

Reply 1

The significance of ASL imaging with two different PLDs has been shown for occlusive cerebrovascular diseases but not for brain tumors. We explained this in the 3rd and 4th paragraphs of the revised Introduction section.

Comment 2

Why did the authors select specific PLDs of 1525 and 2525 ms in their study?

Reply 2

We selected 1525 ms as a standard PLD value according to the vendor’s recommendation and 2525 ms as a longer PLD. We explained this in the 4th paragraph of the revised Introduction section.

The selection of 1525 and 2525 ms was essentially identical to the previous studies (1.5 and 2.5 s in one paper) in occlusive cerebrovascular diseases, as described in the 5th paragraph of the revised Discussion section.

Comment 3

Why didn’t the authors provide ASL imaging parameters gender wise? Was there any sex-dependent variation in these parameters? The authors should provide gender-wise data.

Reply 3

We performed analysis in gender-based subgroups. The concordance between tumor blood flow estimates obtained at PLDs of 1525 and 2525 ms was similarly high in men and women.

In the revised manuscript, we described related matters in 2.3. Data Analysis of the Materials and Methods section, 3rd paragraph of the Results section, and the 9th paragraph of the Discussion section.

Comment 4

The authors consider the data from two different scanner in their study. Can the authors eliminate potential differences due to different scanning machine?

Reply 4

Our observation indicated that ASL imaging with a single PLD is acceptable for both 1.5 T and 3 T scanners.

However, a previous volunteer study indicated a weaker correlation between different scanner models than between different scanners of the same model. The use of the same scanner or the same scanner model is desired for the longitudinal assessment of a given patient. We explained this in the 7th paragraph of the revised Discussion section.

Comment 5

Did the authors find any differences in the ASL assessment depending upon advancement of the tumor?

Reply 5

We added analysis in pathology-based subgroups. The concordance between tumor blood flow estimates obtained at PLDs of 1525 and 2525 ms was confirmed in meningiomas and glioblastomas.

In the revised manuscript, we described related matters in 2.3. Data Analysis of the Materials and Methods, 3rd paragraph of the Results, and the 8th paragraph of the Discussion section.

Comment 6

In the figure 2, error bars are too high!

Reply 6

We agree. The distribution of data are shown in Figures 3-5. The principal results of our study were the concordance in tumor perfusion evaluation between PLDs of 1525 and 2525 ms.

Comment 7

The manuscript should be checked for the grammatical errors and typos.

Reply 7

We checked our revised manuscript for grammatical errors and typos.

Round 2

Reviewer 1 Report

The changes are fine

Reviewer 3 Report

The manuscript is now improved and can be accepted for publication after thorough checking of grammar and spelling.